# Greening the Gas Grid—Evaluation of the Biomethane Injection Potential from Agricultural Residues in Austria

**Bernhard Stürmer** [1,2]

1   Austrian Compost and Biogas Association, Franz-Josefs-Kai 13, 1010 Vienna, Austria;
    stuermer@kompost-biogas.info or bernhard.stuermer@haup.ac.at
2   Institute for Management, Research and Innovation, University College for Agricultural and Environmental
    Education, Angermayergasse 1, 1130 Vienna, Austria

**Abstract:** In order to implement the Paris Climate Agreement, the current Austrian coalition government has included trend-setting targets in its policy statement. "Green gas" plays a key role in this context, as the natural gas grid shall also gradually become renewable. This article analyses the technical biomethane injection potential for agricultural residues based on Integrated Administration and Control System (IACS) data on a municipal level. While a technical biogas potential of 16.2 $TWh_{CH4}$ from catch crops, farm manure, straw and beet leaves is available, only about half of it can be fed into the gas grid because of technical and economic reasons. Austria's biomethane injection potential of 7.4 $TWh_{CH4}$ is mainly produced in arable farming regions. In order to further increase this potential, the investment costs of biogas upgrading plants must be reduced, the use of biogenic waste and energy crops must be further promoted and an investor-friendly legal framework must be created.

**Keywords:** biogas; biomethane potential; green gas; agricultural residues

## 1. Introduction

The energy sector was listed as the main greenhouse gas emitter in the first IPPC (Intergovernmental Panel on Climate Change) report in 1990 [1]. Achieving the Paris climate goal of curbing global warming to a maximum increase of 1.5 °C, makes substantial efforts necessary. These also hold risks, especially for regions with a high dependency on fossil fuels. Therefore, forward-looking policies that promote diversification of the energy and economic sectors need to be developed to ensure income generation and job creation. Due to its potential, the use of bioenergy can make a contribution as a substitute for fossil fuels in all sectors [2].

Following the Paris Climate Agreement [3] and the European Green Deal [4], the current Austrian government integrated the chapter "Climate Protection & Energy" in its policy statement, which defines climate neutrality by 2040 at the latest as its overarching goal. This is to be achieved by increased energy efficiency, a phase-out plan for fossil fuels in heating and the expansion of renewable energy sources. To finance the conversion of the energy system, the government plans to leverage private money and to prepare an eco-social tax reform [5].

To promote renewable energy sources, measures shall be taken on the one hand to cover 100% of the electricity consumption on the balance sheet from green electricity by 2030. On the other hand, an expansion and support scheme for "green gas" shall be introduced, which shall lead to an annual injection capacity of 5 $TWh_{CH4}$ into the gas grid by 2030. "Green gas" includes biomethane, green hydrogen and synthetic gas based on renewable electricity [5]. Feeding green gas into the gas

grid is also a top priority, as its transmission and storage serve to ensure security of supply in the gas and electricity sector, reduces emissions in the transport sector and supply heat to densely populated urban areas [6–10].

Biogas technology can thus contribute to both areas [11,12]. Biogas plants have undergone considerable expansion in Europe over the past 15 years. National laws and regulations have had a considerable influence on these expansion rates. While in Germany, for example, there were two major waves based on the Renewable Energy Sources Act in 2004 and especially in 2009 [13,14], around 80% of all currently active Austrian plants went into operation between 2003 and 2007 based on the Green Electricity Act 2002 [15]. National laws and ordinances have established systems that guarantee fixed feed-in tariffs or feed-in premiums or allocate tradable certificates to the operators of renewable electricity generation plants [16]. This support is intended to cover the full costs of electricity generation, as selling electricity at marginal cost on the electric power exchange would not lead to an expansion of (renewable) electricity capacities [17]. In Europe, several countries rely on the production of biomethane, next to the electricity production, based on anaerobic digestion. Especially in Germany, Great Britain, France and Sweden, the numbers of biomethane injection plants raise in the last decade. Nearly 23 TWh$_{CH4}$ of biomethane was produced in 2018 [18]. Austrian biogas plants produce currently about 560 GWh of green electricity and about 150 GWh of biomethane annually [19,20].

In addition to the expansion of green electricity capacities, an added political focus is now to be placed on the injection of biomethane. Whatever the package of measures will ultimately look like, the question still arises whether such quantities of biogas or biomethane can be supplied. Since the impact of using agricultural raw materials on agricultural prices have been widely discussed, the use of biogenic waste and agricultural residues is favored [21]. Two studies have recently been carried out to estimate Austria's biomethane injection potential. While Dißauer et al. [22] also included woody biomass for the production of green gas (wood gas), Lindorfer et al. [23] also analyzed the synthetic methane potential from power-to-gas plants in addition to the biomethane injection potential. Both studies particularly considered straw, beet leaves and farm manure as resources for the agricultural sector. However, the results of the quantity estimates differ significantly. Furthermore, the location of the gas network was not taken into account in these calculations.

This article makes an analysis based on the agricultural production data at municipal level. In addition to evaluate the amounts of farm manure and straw, it also discusses catch crops as another raw material source. Catch crops hold further potential for the production of biomethane, as they are well suited as feedstock, especially due to their dry matter content in combination with straw [24]. Moreover, technical limitations are taken into account and transport distances for the delivery of raw materials are estimated, which cannot be specified in the scope of a national or federal state analysis.

## 2. Materials and Methods

In order to derive the biogas potential, this article uses Integrated Administration and Control System (IACS) data at municipal level. The IACS data enable information about production of each farm and are used for direct payment claims [25]. In addition, municipalities with a gas network connection were identified in order to determine whether the biogas plants could also become biomethane injection plants or whether electricity production represents the only way of biogas usage. The "Austrian Gas Grid Management AG" made an unpublished data set of the municipalities, which have a gas connection, available [26]. These data are supplemented by regional data of Statistic Austria [27].

The IACS database holds information about the grain, maize, rapeseed and beet areas. The coefficients listed in Table 1 were used to estimate the resulting yield as well as the specific methane yield. According to the Federal Ministry for Sustainability and Tourism [28], catch crops were cultivated on 265,000 ha (annual active cultivation and area-wide greening with catch crops, 6 different cultivation and conversion variants). Moreover, the "evergreen" system (area-wide greening of at least 85% of arable land at any point in time) was used on close to 200,000 ha. For the present potential assessment, it is assumed that all grain, grain maize and grain legume areas can subsequently

be cultivated with a greening variant. Areas for rapeseed production are excluded, as the time of cultivation in August excludes catch crops. Overall, this approach covers a total area of 743,000 ha.

**Table 1.** Assumptions regarding yield (in tons of dry matter per hectare) and specific methane yield for straw, beet leaves and catch crops (in cubic meters of methane per ton of dry matter) [24,29–32].

|  | Yield [$t_{DM}$ $ha^{-1}$] | Specific Methane Yield [$m^3_{CH4}$ $t_{DM}^{-1}$] |
|---|---|---|
| Cereal straw | 4.2 | 174 |
| Maize stover | 5.0 | 308 |
| Rapeseed straw | 2.8 | 188 |
| Beet leaves | 3.6 | 275 |
| Catch crops | 2.6 | 280 |

In order to derive the amount of farm manure, information available from the IACS database regarding the number of animals in the categories cattle, pigs, poultry, sheep and goats as well as other farm animals (especially horses) was supplemented by data from the Federal Ministry of Agriculture, Forestry, Environment and Water Management [33]. Because of differences at animal categories and age of animals per municipality, Table 2 shows the mean value and the standard deviation for the amount of farm manure per livestock unit. The specific methane yields were based on the Association for Technology and Structures in Agriculture and the Bavarian State Research Center for Agriculture [30,34].

**Table 2.** Mean value and standard deviation for the amount of farm manure in tons of fresh matter per livestock unit (LU) and the specific methane yield in cubic meters of methane per ton of dry matter [30,33,34].

|  | Mean Value [$t_{FM}$ per LU *] | Standard Deviation [$t_{FM}$ per LU] | Specific Methane Yield [$m^3_{CH4}$ $t_{DM}^{-1}$] |
|---|---|---|---|
| Cattle | 18.5 | 1.53 | 167 |
| Pigs | 5.5 | 1.03 | 186 |
| Poultry | 7.4 | 1.31 | 225 |
| Sheep and Goats | 6.7 | 0.81 | 330 |
| Other livestock | 6.3 | 3.77 | 125 |

* 1 LU ≈ 500 kg live weight (cf. [35]).

The biomethane injection potential was assessed in three successive steps:

1. The first stage is the "theoretical biogas potential" from agricultural residues. No restrictions are placed on land use or the amount of agricultural manure in this stage.

2. The second stage, the "technical biogas potential" includes restrictions to consider technical limitations. Therefore, minimum amounts for catch crops (≥300 ha per municipality) and farm manure (≥12,000 $t_{FM}$ per municipality) are defined to take logistic limitations into account. Moreover, 300 ha of catch crops or 12,000 t of farm manure are necessary to meet the methane demand of a biogas plant with an electric capacity of 100 kW and 7200 full-load hours (200,000 $m^3_{CH4}$). Sugar beet leaves are harvested if sufficient catch crops and/or farm manure is available in a community. In addition, restrictions are placed on the use of straw as feedstock. Since a dry matter content of over 9% in the fermenter requires a special agitator technology [36] and the amount of water or recyclate that can be added is limited by the fermenter volume, the dry matter content of the feedstock mixture is limited to 30%. With a dry matter content of 30%, the necessary amount of water corresponds to the amount of feedstock to achieve a dry matter content of 15% in the fermenter. More water or recyclate as well as biological degradation reduce the dry matter content in the fermenter to below 15%. The amount of straw that can be added, which has a dry matter content of around 86%, is therefore corrected for technical reasons.

3. In the third stage, the biomethane injection potential was calculated. In addition to the limitations regarding the technical potential, assumptions for the derivation of the biomethane potential are included: in communities where a gas supply is available, an annual minimum quantity of 1,200,000 $m^3_{CH4}$ is required to operate a biomethane injection plant with 150 $m^3_{CH4}$ $h^{-1}$ (8000 full load hours), which is a common feed-in capacity in Austria [37]. In municipalities where this requirement is not met, or that do not have a gas supply, the resulting gas volume is processed in biogas plants for combined heat and power generation.

According to the Austrian "Gas System Usage Fee Ordinance" the factor 11.33 kWh $m^{-3}_{CH4}$ was used to convert the biomethane quantities into energy [38].

To derive the average haul distance, the number of biogas plants was determined in a first step based on the technical potential. The following size classes were established for this purpose:

- Biomethane injection plant: 150 to 300 $m^3_{CH4}h^{-1}$
- Electricity generation plant: up to 150 kW electric
- Electricity generation plant: 150 to 400 kW electric
- Electricity generation plant: 400 to 750 kW electric
- Electricity generation plant: 750 to 1300 kW electric

The existing potential was sorted into these size classes and the number of plants per municipality were determined. A capacity utilization of 8000 full-load hours was assumed to estimate the number of electricity generation plants. For example, if a region can supply a biogas plant with a capacity of 220 kW with feedstock, the plant is sorted into the size class 150 to 400 kW. Regions with a technical potential of more than 1300 kW were divided into several plants of the size class 400 to 750 and 750 to 1300 kW respectively.

Following Overend [39], it was assumed that the biogas or biomethane injection plant is located in the center of a circle. The area is given by the total available area of a municipality. In the case of several plants, the area was divided accordingly by the number of plants. According to Overend, the average haul distance is determined from the 2/3 radius of the circle, taking a tortuosity factor into account. This was assumed to be 1.33, which represents the difference between the linear distance and road distance.

## 3. Results

### 3.1. Theoretical and Technical Biogas Potential

Assessment of the data showed that Austria produces around 31.6 million $t_{FM}$ of farm manure, which corresponds to the estimation of Zethner and Süßenbacher [40]. In addition, an area for the cultivation of almost 8.8 million $t_{FM}$ of catch crops is available. Rapeseed and cereal straw as well as maize stover are available in the amount of around 4.0 million $t_{FM}$. Around 740,000 $t_{FM}$ of sugar beet leaves can be harvested from 31,000 ha of sugar beet. The quantity limit for estimating the technical biogas potential affects catch crops (−12%) and farm manure (−16%). This is accompanied by a reduction in the possible use of straw. The straw input is reduced by 37% compared to the theoretical potential (Table 3).

**Table 3.** Theoretical and technical potential compared to current potential studies in Austria.

| | Theoretical Biogas Potential | | Technical Biogas Potential | | Dißauer et al. [22] | Lindorfer et al. [23] |
|---|---|---|---|---|---|---|
| | $10^6$ $t_{DM}$ | TWh$_{CH4}$ | $10^6$ $t_{DM}$ | TWh$_{CH4}$ | $10^6$ $t_{DM}$ | $10^6$ $t_{DM}$ |
| Catch crops | 1.9 | 6.1 | 1.7 | 5.4 | − | − |
| Straw | 3.5 | 8.4 | 2.2 | 5.3 | 4.0 | 1.7 |
| Beet leaves | 0.1 | 0.3 | 0.1 | 0.3 | 0.25 | 0.1 |
| Farm manure | 3.1 | 6.2 | 2.6 | 5.1 | 8.5 | 3.9 |
| Sum | 8.7 | 21.0 | 6.6 | 16.2 | 12.75 | 6.1 |

In comparison to the studies of Dißauer et al. [22] and Lindorfer et al. [23], the present results can be classified in between the theoretical and technical potential shown in this article, as the available amounts were assumed differently in both studies. As can be seen in Table 3, the straw and sugar beet leaf potentials are rather in the range of the Lindorfer study. The estimation for the amount of farm manure was most likely too high in both studies.

In terms of spatial distribution, those municipalities that are part of Austria's arable farming regions have a high biogas production potential from agricultural residues (Figure 1). Due to livestock farming, grassland areas also hold a considerable biogas production potential, though the specific production potential is lower than in arable farming areas.

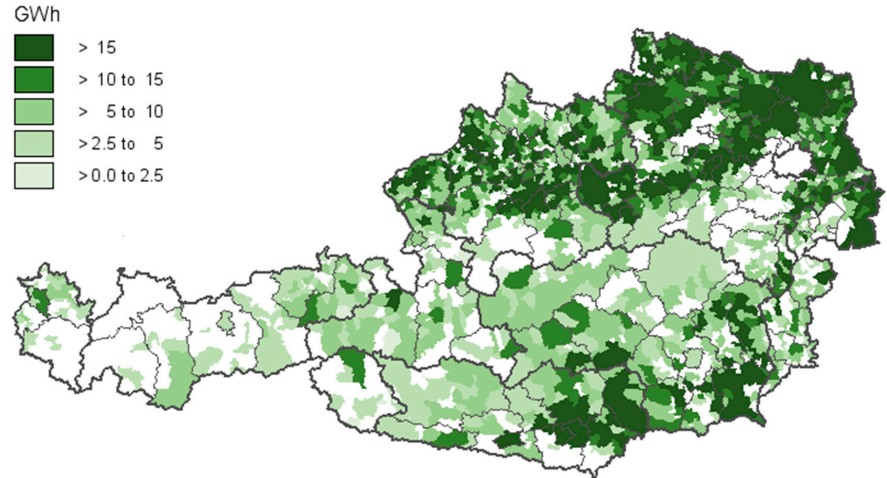

**Figure 1.** Spatial distribution of the technical biogas potential (own analysis).

### 3.2. Biomethane Injection Potential

To derive the biomethane injection potential, only those municipalities with a gas network connection were considered. Moreover, the annual minimum quantity of 1,200,000 $m^3_{CH4}$ to operate a 150 $m^3_{CH4}h^{-1}$ biomethane injection plant was considered. In Figure 2 those municipalities, where the existence of a biomethane injection plant is probable, are marked in color. Overall, 284 municipalities are equipped to inject 7,4 $TWh_{CH4}$ or 46% of the technical biogas potential into the grid. The largest biomethane injection potential of 3.8 $TWh_{CH4}$ is found in Lower Austria, followed by Upper Austria with 2.2 $TWh_{CH4}$ (both in northern Austria). The southeastern federal states of Styria, Burgenland and Carinthia show a respective potential of 0.6, 0.4 and 0.3 $TWh_{CH4}$, respectively.

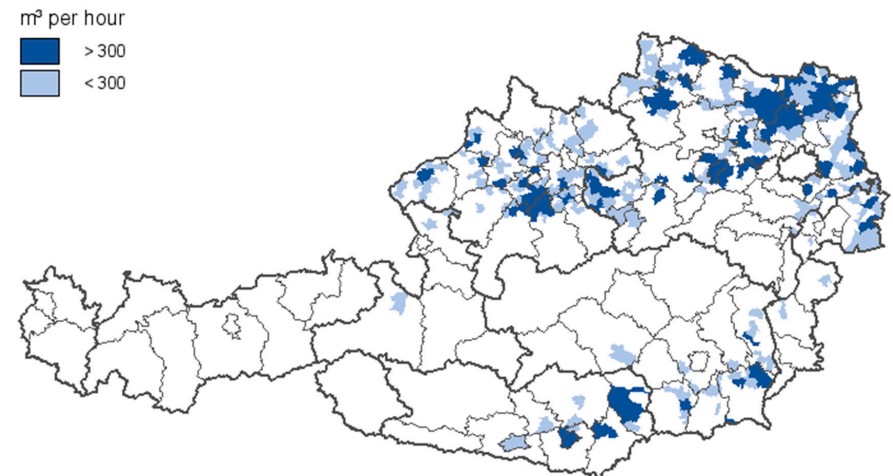

**Figure 2.** Regional distribution of the possible biomethane injection potential (own analysis).

Around 370 MW electrical capacity can be installed in those 1032 municipalities, where the technical potential is converted into electricity. In 779 municipalities, the technical potential does not meet the requirements for an electricity generation plant with an electric capacity of 100 kW.

### 3.3. Average Haul Distance

Calculation of the biomethane injection potential showed that, overall, 393 biomethane injection plants can be operated in 284 municipalities. In those 1032 municipalities, where the potential is sufficient for electricity generation plants, 128 plants with a capacity of up to 150 kW (Ø 127 kW), 620 plants with a capacity of up to 400 kW (Ø 249 kW), 263 plants with a capacity of up to 750 kW (Ø 518 kW) and 63 plants with a capacity of up to 1300 kW (Ø 981 kW) can be operated. The average haul distance was calculated by equally distributing the municipal area among the number of plants within the municipality. This showed that 92% of all plants have a maximum haul distance of 5 km (Figure 3). On average, the haul distance for biomethane injection plants is at 2.8 km slightly shorter than for electricity generation plants at 3.1 km.

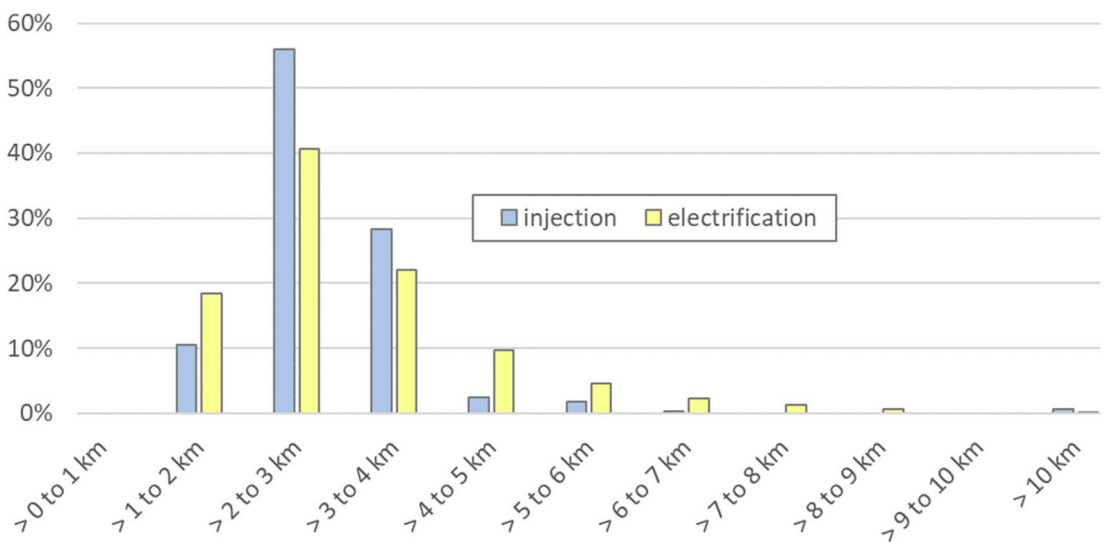

**Figure 3.** Frequency distribution of average haul distance (own analysis).

### 3.4. Sensitivity Analysis

For the analysis of the second potential level, restrictions were included to take logistical limitations into account. This was on the one hand a minimum amount of farm manure ($\geq$12,000 $t_{FM}$ per municipality), on the other hand a minimum area for the cultivation of catch crops ($\geq$300 ha per municipality). Figure 4 shows the extent to which these limitations affect the potential of biomethane injection and biogas electrification. If the minimum amount of farm manure is set at 6000 (−50%) or 18,000 $t_{FM}$ (+50%), the technical potential changes by +8.4% and −7.9%, respectively. If the minimum area for catch crops is reduced to 150 ha per municipality −50%), the technical potential increases by 3.8%, while setting the minimum area to 450 ha (+50%) reduces the potential by 4.5%.

However, the critical variable for assessing the achievable biomethane injection potential is the feed-in capacity, as shown in Figure 5. Assuming a common feed-in capacity of 150 m$^3$ of biomethane per hour or more, about 46% of the technical potential can be fed into the gas grid. If this limitation is set to 30 m$^3_{CH4}$h$^{-1}$, the share of the technical potential increases to 67%. If the requirement is set to at least 200 m$^3_{CH4}$h$^{-1}$, the share for biomethane injection is reduced to 37% of the technical biogas potential.

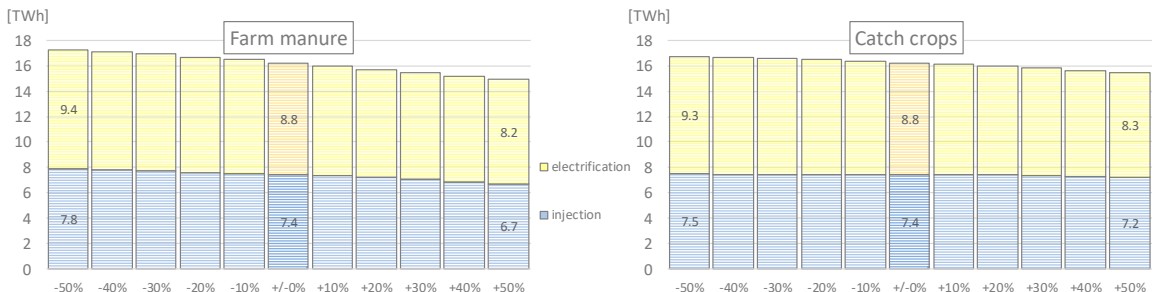

**Figure 4.** Changes in the biomethane injection and electrification potential based on changes in the minimum amounts for farm manure (**left**) and catch crops (**right**) (own analysis).

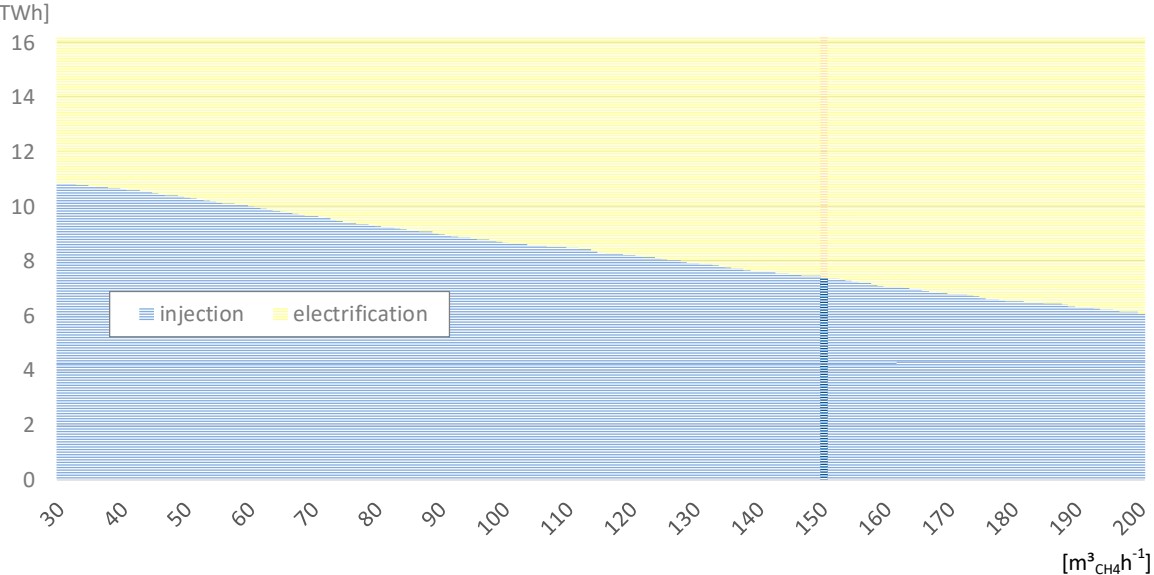

**Figure 5.** Change in biomethane injection potential based on changes in the minimum feed-in capacity (own analysis).

## 4. Discussion

This article analyzed the technical biomethane injection potential based on agricultural data at the municipal level. In addition to farm manure and straw, sugar beet leaves and potential catch crop areas were also considered. The biomethane injection potential of 7.4 $TWh_{CH4}$ from agricultural residues is mainly produced in Austria's arable farming regions. On the one hand, this is due to the fact that the necessary amount of feedstock for the operation of biomethane injection plants is more easily achievable in these regions. On the other hand, this is also due to the fact that these regions have a better gas supply.

The current government's aim of injecting 5 $TWh_{CH4}$ of "green gas" [5] into the gas grid can be achieved via the utilization of agricultural residues. However, at around 95 $TWh_{CH4}$, current gross domestic consumption is still well above the current volume target. Moreover, with regards to the goal of no longer emitting additional fossil $CO_2$ by 2040 at the latest, the question arises as to how the current demand for natural gas can be met by alternative sources. Therefore, the focus needs to be directed two-ways. On the on hand, the use of green gas needs to be promoted. Gas is currently particularly consumed by gas-fired power plants for electricity generation, by the non-energy sector (e.g., the chemical industry) and by households for cooking and heating purposes [41]. On the other hand, the additional availability of green gases must be analyzed. The studies of Dißauer et al. and Lindorfer et al. [22,23] therefore also considered woody biomass for the production of wood gas as well as the potential of synthetic methane from power-to-gas plants.

In the field of biogas technology, the use of energy crops should also be reconsidered. If energy crops, such as maize, are included in the calculations, biomethane injection benefits disproportionately compared to electricity generation. The potential increases are about three times higher for biomethane injection than for electricity generation. Therefore, potential assessments should take feedstock quantities from grassland and arable farming into account. Energy crops that are not suitable for food or feed production (e.g., mycotoxin-contaminated cereals, drought-damaged maize, etc.) also hold potential. Utilization of these energy crops would also reduce the average haul distance, which at 3 km is higher than in the analyses of Walla and Schneeberger [42] or Stürmer et al. [43] at Austrian circumstances. The use of energy crops would also enable the operation of biogas plants in those regions where, according to the present analysis, biogas production is currently not possible.

Structural change, especially in livestock farming, is to be expected in the coming years and decades [28]. This only has minor implications for the biogas potential assessment, as on the one hand, farms that give up livestock farming are compensated by larger farms. On the other hand, land potentials are released if they are no longer used for livestock farming.

The assessment of the Austrian biomethane potential from the waste sector is somewhat vague because of lack of regionalized data. Digestion of predestined waste (e.g., waste from the gastronomy, food, feed and beverage industry, etc.) could also increase the potential, as shown per example in Feiz et al. [44]. However, these data are currently not available in Austria in the required regional breakdown and can, therefore, only be estimated on a general level. Yet, high-energy consumption by these industries suggests a gas connection. Because of this physical proximity, new biomethane injection plants based on biogenic waste can be expected.

Future technical developments of biogas technology and especially of biogas upgrading facilities is also relevant, as a decrease in the specific feed-in capacity is necessary to increase biomethane injection quantities. The Austrian biogas sector is marked by relatively small plants. Biomethane injection capacities of up to 150 $m^3_{CH4}h^{-1}$ are the most probable plant sizes [37]. For economic reasons, it is therefore necessary to reduce the specific investment costs to such an extent that smaller plants also become competitive [45,46]. This way, the available resources can be used to increase the share for biomethane production.

However, in order to physically inject biomethane in to the gas grid, the economic framework conditions are decisive (see e.g., [37]). Private investors, whom the Federal Government would like to approach more intensively, will only make the necessary investments, if the investment risk is manageable. In addition to sufficiently high annual revenues, attention should also be paid to sufficiently long contract terms in order to keep the average annual capital costs low. Moreover, it is necessary to determine which of the systemically established regulations artificially increase the cost of biomethane production. For example, the required Wobbe index cannot be achieved by biomethane because pure methane has a lower calorific value than the methane-ethane-propane-butane mixture of natural gas [47]. The Wobbe Index is not allowed to be less than the supplied natural gas. Therefore, gas grid operators set requirements for biomethane injection, which can only be reached by blending propane gas. Lowering these requirements for biomethane would reduce the additional costs caused by the need to blend propane gas [48]. Biomethane injection plants also quickly reach technical limits when injecting into local grids, as the amount of biomethane produced cannot be absorbed by the gas grid due to insufficient offtake. The necessary recompression plants to reach higher gas network levels are currently being constructed at the cost of the biomethane producer. If the biomethane injection plant is considered as part of network infrastructure maintenance, costs for recompression could also be borne by the gas network operator and written off over a longer time period.

Favorable legal framework conditions leading to an expansion of biogas and biomethane production are expected to have further effects. If the potential is fully exploited, as shown in this article, a total of 1316 rural municipalities will benefit from new businesses and new jobs. The biogas sector is also very input-intensive and closely linked to the surrounding economy. According to

Koller [49], the domestic share of production and value added exceeds 85%, which in turn benefits existing companies in rural areas.

**Funding:** This research received no external funding.

**Conflicts of Interest:** The author declares no conflict of interest.

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
