# Peer review of "Greening the Gas Grid—Evaluation of the Biomethane Injection Potential from Agricultural Residues in Austria"

_processes, doi:10.3390/pr8050630_

Round 1

Reviewer 1 Report

The evaluation of resources for biogas and biomethane production is an important issue for further design of programs and actions plans. Nevertheless, this is a process affected by multiple variables and uncertainties. Paper will gain in novelty if methodology applied is explained in a manner that could be applied elsewhere, justifying and discussing constrains, and including the management of uncertainties.

Abstract

Line 17: add reasons, in short, explaining why about the half can be fed into the gas grid: economic, environmental,… constrains?

What is the meaning of IACS? Probably the explanation is not necessary in the abstract section, but in the text.

Introduction

Based on EBA (European Biogas Association) Statistical Report 2018, Austria has 423 biogas plants producing 565 GWh electricity and 15 biomethane plants producing 146 GWh, data from 2016. Nothing is said in the introduction about these already operating plants, and whether some of these plants are fed with raw materials included in the resources inventory of the current study. The state of art in Austria should be described, and the current policies and energy prices enabling the operation of the indicated plants, in order to draw a useful picture for readers.

Materials and methods

Remember to describe the meaning of IACS (Line 74).

Table 1: different catch crops are possible. Indicate what the preferable catch crops are in Austria, considering climatology or other agricultural issues, and the corresponding methane yields.

All the yields (crop yields and methane yields) indicated in Tables 1 and 2 should be characterized by its confidence interval or standard deviation, as was done for manure production. If this is not possible, consider to add a sensitivity analysis of the variation of these yields around the mean value.  Owing to the number of factors affecting the quality and the methane yield, organic waste inventories intended for biogas plants planning should be characterized by an uncertainty degree, or by intervals as was done by Lorenz et al. (2013, doi: https://doi.org/10.1016/j.wasman.2013.06.018 ) in a study of energy potential from biogas in the UE-27.

FM and DM is thought to be fresh matter and dry matter, respectively, but not indicated. Please, indicate the meaning.

Table 2: GVE is supposed to be some unit related to livestock unit, but not indicated. Reference some work where livestock units are defined and the corresponding weights for each livestock type.

Justify the constraints adopted: 150 m3 CH4·h-1, minimum amounts for catch crops, etc. Are these economic constrains?

Indicate the electrical efficiency used for deriving the electrical power of the biogas plants for this purpose. Are those of Walla and Schneeberger (2008)?

It is not indicated whether the thermal and electrical energy required for the operation of the biogas and biomethane plants have been subtracted from the energy potential estimated. Has it been done? If not, why not?

Results and discussion

Lines 271-272: this sentence is unclear (disproporcionate?)

Lines 286-288: this sentence is unclear, since the Wobbe index for pure methane is 48 (LWI) to 53.4 (HWI), values that fit the Austrian regulations. Probably you mean that current regulations obligate to a minimum CH4 content in biomethane? Explain in more detail.

Author Response

Thank you very much for the valuable notes!

Reviewer 2 Report

Biomethane production technology is becoming increasingly popular all over the world. Biomethane can be used as a substitute for natural gas. The authors presented "Evaluation of the biomethane injection potential from agricultural residues in Austria”. The work is written very well. Authors should explain what abbreviations mean (e.g. IACS, GVE). The quality of Figures 1 and 2 should be better. Please describe the biomethane (and biogas) market in the EU more widely. Please quote more articles from the journal.

Author Response

(The authors gave the same response as above.)

Round 2

Reviewer 1 Report

The questions have been answered adequately and the corrections to the manuscripts correspond to these answers.

The answer to the question about Wobbe index is satisfactory, but I think the corresponding sentence in the manuscript could be improved adding some of the sentences of the answer, in order to clarify for readers. I suggest the following, as example, (lines 604-606):

“For example, the required Wobbe index cannot be achieved by biomethane because pure methane has a lower calorific value than the methane-ethane-propane-butane mixture of natural gas, and the Wobbe Index is not allowed to be less than the supplied natural gas. Therefore, gas grid operators set requirements for biomethane injection which can only be reached by blending propane gas.”

Author Response

Thank you very much for the suggestion. Sentence is implemented.